# Seed-induced acceleration of amyloid-β mediated neurotoxicity in vivo

Ramona F. Sowade[1] & Thomas R. Jahn [1,2]

Seeded propagation of amyloid-beta (Aβ) pathology is suggested to contribute to the progression of Alzheimer's disease. Local overproduction of aggregation-prone Aβ variants could explain the focal initiation of a seeding cascade that subsequently triggers widespread pathology. Several animal models support this seeding concept by demonstrating accelerated Aβ deposition following inoculation with Aβ-containing homogenates, however its role in progressive neurodegeneration remains unclear. Here, we present a non-invasive approach to study Aβ seeding processes in vivo using *Drosophila* models. We show that small amounts of aggregation-competent Aβ$_{42}$ seeds, generated in selected neuronal clusters, can induce the deposition of the pan-neuronally expressed and otherwise soluble Aβ$_{40}$. Moreover, our models visualize the accelerated formation and propagation of amyloid pathology throughout the brain, which correlates with severe neurotoxicity. Taken together, these in vivo models provide mechanistic insights into disease-related processes and represent versatile genetic tools to determine novel modifiers of the Aβ seeding cascade.

[1] Proteostasis in Neurodegenerative Disease (B180), German Cancer Research Center, Schaller Research Group at the University of Heidelberg and DKFZ, Heidelberg 69120, Germany. [2] Present address: AbbVie Deutschland GmbH & Co. KG, Ludwigshafen 67061, Germany. Correspondence and requests for materials should be addressed to T.R.J. (email: thomas.r.jahn@abbvie.com)

The most prevalent form of dementia, Alzheimer's Disease (AD), is characterized by the misfolding and accumulation of the amyloid-β (Aβ) peptide, resulting in the formation of extracellular senile plaques as a characteristic pathological hallmark[1–3]. Early changes in Aβ proteostasis leading to increased aggregation of this peptide have been indicated as initial steps in the development of AD, as they may cause downstream pathological lesions, such as neurofibrillary tangles and the initiation of neuroinflammatory processes[4–6]. Hence, detailed insights into the molecular mechanisms underlying Aβ aggregation are crucial to understand the etiology of AD and thereby to identify novel therapeutic targets.

Post-mortem studies suggest that Aβ aggregates do not form stochastically throughout the brain but rather deposit in a stereotypical manner with lesions first found in the neocortex, then in the allocortex and at later stages also in subcortical regions[7–9]. However, the relevance for this characteristic progression of pathology in disease staging remains unclear, as plaque load correlates only to a limited extent with disease severity[10]. Interestingly, the amount of total Aβ was determined as a more reliable measure to estimate disease progression[11, 12], indicating the complex relationship between aggregation status, propagation of pathology and neurotoxicity, which still remains a major challenge in AD research.

In recent years, several studies have shed light on the molecular mechanisms leading to the progression of Aβ pathology. Current results indicate the intriguing role of templated protein misfolding, usually referred to as seeding, as a crucial mechanism in the initiation and propagation of Aβ deposition in the brain[9, 13–16]. According to this concept, a small portion of aggregated peptide acts as a template to induce misfolding and aggregation of the normally soluble protein[17]. This concept was extensively explored using in vitro studies, where the formation of amyloid fibrils can be accelerated by the addition of pre-formed seeds at the beginning of the aggregation process[18–20]. Thereby, primary nucleation during the lag phase of fibril formation, i.e., the assembly of monomers into oligomers and fibrils, can be shortened proportional to the amount of introduced seeds[21, 22]. Primary nucleation, is a rather slow process, which alone cannot account for the steep aggregation curve observed in in vitro studies[23, 24]. Thus, secondary nucleation mechanisms have been suggested as the major driving force in progressive protein aggregation in vitro and in vivo[22, 23]. These mechanisms might include breakage of existing fibrils giving rise to an increasing

number of fibril ends being available for further monomer addition. With an excessive amount of these seeds, the rate-limiting factor for aggregation becomes the availability of soluble peptide monomers as building blocks to nascent fibrils[22, 23, 25]. In the case of AD, the local overproduction of aggregation-prone Aβ$_{42}$ variants might give rise to such potent protein seeds, thereby inducing the aggregation of abundant and normally soluble Aβ variants, such as the shorter Aβ$_{38}$ and Aβ$_{40}$ variants. A confined generation of seeds could, for example, result from the selective vulnerability of certain neuron types to changes in proteostasis[26]. Another source might be the recently confirmed occurrence of somatic mosaic mutations in patients' brains, which could lead to the restricted production of fast-aggregating Aβ variants by small neuronal clusters subsequently initiating the seeding cascade[27, 28].

Several animal models support this seeding concept by showing accelerated Aβ pathology in host organisms after intracerebral injection with Aβ aggregate-containing brain homogenate[15, 16, 29–32]. This effect can be prevented by immunodepletion of Aβ from the injected extracts, thereby supporting the direct role of Aβ as the seeding agent in this process[16]. Seeding strongly correlates with the applied Aβ concentrations and the time period after injection, suggesting a direct nucleation mechanism behind the accelerated Aβ pathology[16, 30, 32]. The rate and characteristics of induced Aβ pathology further depend on the cerebral area in which the seeding-competent material is injected, with strongest deposition in the hippocampus and entorhinal cortex, i.e., in regions that are also severely affected in transgenic mice under normal aging conditions[30]. This finding indicates that the starting point of the seeding cascade is crucial for its course and thereby supports the idea of a selective vulnerability of certain brain regions. Notably, injection of seeding-competent material into wild-type mice did not cause increased Aβ deposition, showing that induction of pathology is not only dependent on the presence of seeds, but also on the amount of the soluble target peptide[16]. Overall, results from rodent models suggest small amounts of Aβ seeds as potent drivers of Aβ pathology in vivo. While these models are based on the exogenous and invasive injection of seeding-competent material into the host organism, it would be advantageous to explore whether a similar seeding effect can also be observed in vivo in an intact neuronal system, and how such endogenous Aβ seeding processes may contribute to neurotoxicity.

Here, we describe *Drosophila* in vivo models, where both the seed and the target protein are fully genetically encoded, allowing mechanistic conclusions on the Aβ seeding processes in an intact neuronal network. Our results show that small amounts of disease-related and fast-aggregating Aβ species (seeds) induce the deposition of an abundant and normally soluble Aβ variant (target), thereby initiating the progression of Aβ pathology. Importantly, we also demonstrate that the accelerated formation of Aβ deposits is attended by a reduction in fly survival pointing to severe neurotoxic effects. This proof-of-concept study provides direct evidence for a link between Aβ-seeding mechanisms and neurotoxicity in vivo. The described novel invertebrate models, therefore, represent a powerful system for the mechanistic characterization of seeding processes as driving force in disease progression.

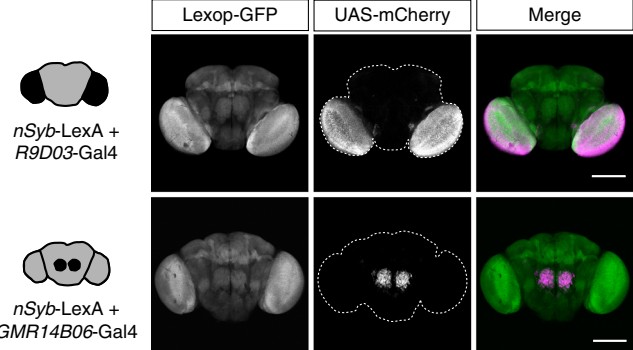

**Fig. 1** Dual expression systems allow simultaneous but independent expression of two transgenes in the fly brain. Confocal microscopy images of adult *Drosophila* brains, showing the expression of GFP (*green*) under control of the pan-neuronal *nSyb*-LexA promoter, and restricted mCherry (*magenta*) expression driven by two independent Gal4-promoters (*R9D03*-Gal4 and *GMR14B06*-Gal4). Merge images are pseudocolored. *Scale bars*, 200 μm

## Results

**Generation of dual-expression *Drosophila* lines.** To investigate the effect of seeded Aβ aggregation in a non-invasive in vivo model, we established a set of novel *Drosophila* lines by combining the well-established transgene expression systems Gal4/UAS[33, 34] and LexA/LexAop[35, 36] in order to

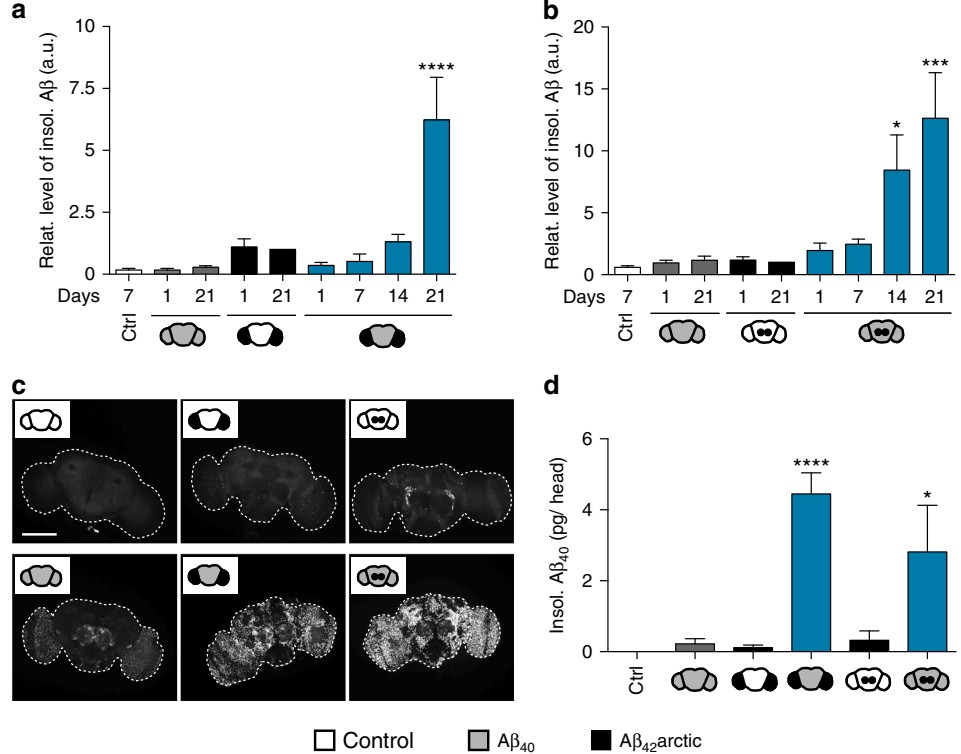

**Fig. 2** Seeding by $A\beta_{42}$ results in the deposition of $A\beta_{40}$ in the insoluble fraction. **a, b** Quantification of the amount of insoluble (insol.) $A\beta$ determined by western blotting. The $A\beta_{42}$arctic seeds were expressed either in a subset of neurons in the optic lobes (**a**, *R9D03*-Gal4) or in a small neuronal cluster in the central brain (**b**, *GMR14B06*-Gal4). $A\beta$ levels were normalized to $A\beta_{42}$arctic alone at day 21. Control (Ctrl) represents driver line only. *Error bars* indicate s.e.m., $n = 3$ independent biological replicates, one-way ANOVA (Dunnett's multiple comparisons test) in comparison to d1 $A\beta_{42}$arctic only, *$P = 0.0256$, ***$P = 0.0005$, ****$P = 0.0001$. Apparent low signals for $A\beta$ in Ctrl flies are caused by unspecific background on western blots. Glyoxalase 1 was used as a loading control. **c** *Drosophila* brains were dissected at day 21 and stained with the monoclonal $A\beta$ antibody 6E10. The sketch in the upper left corner of each image represents the area of transgene expression. *Scale bar*, 150 μm. **d** ECL detection of insoluble $A\beta_{40}$ in head extracts of *Drosophila* aged 21 days (*error bars*, s.e.m., $n = 3$ independent biological replicates, one-way ANOVA (Dunnett's multiple comparisons test) in comparison to $A\beta_{40}$ only, *$P = 0.0112$, ****$P = 0.0001$)

express two $A\beta$ variants simultaneously, but independently, in the same brain. For the identification of suitable neuronal clusters for locally restricted transgene expression, we screened a neuronal Gal4-driver line collection with distinct expression patterns[37, 38]. To provide insights into the role of specific cerebral areas as well as the amount of required seeding material, we subsequently decided to focus our studies on two different Gal4 lines for further analysis, namely *R9D03*-Gal4 and *GMR14B06*-Gal4. As illustrated in Fig. 1, both driver lines are specific to a selected number of neurons, but distinct in the localization of expression, as well as the number of neurons expressing the transgene. While *R9D03*-Gal4 transgene expression is distributed over the optic lobes, the *GMR14B06*-Gal4 driver is restricted to a small subset of neurons in the central brain (Fig. 1). As such, we are able to explore whether the seeding effect depends on the cerebral region in which the seed is expressed and on the overall amount of generated seed material. By combining these Gal4 driver lines with the *nSyb*-LexA driver for the independent, pan-neuronal expression of a second transgene, we generated two alternative dual expression systems. As illustrated by the distinct expression of the two fluorescent protein variants mCherry and GFP (Fig. 1), this setup allows the differential expression of one protein variant in a small neuronal cluster, while a second protein can be independently expressed in all neurons. Expression of these Gal4 driver lines overlaps with the pan-neuronal expression by *nSyb*-LexA to ensure that seed and target peptides can interact with each other, as it is unclear whether $A\beta$ seeds spread

throughout the brain autonomously. This expression profile mimics the physiological scenario, where different $A\beta$ variants are produced by the same cell and within the same brain regions. Establishing such a setup allows the analysis of in vivo seeding mechanisms in a non-invasive and genetically tractable system. This advancement over invasive seeding models enables us to study seeding mechanisms directly in vivo, and to link these processes with the progression of neurotoxicity in closed biological models.

**$A\beta_{42}$ seeds initiate the deposition of $A\beta_{40}$.** We subsequently investigated whether small amounts of a fast-aggregating $A\beta$ variant can seed aggregation of an abundant and normally soluble $A\beta$ species in an intact neuronal system. Here, we used two human $A\beta$ peptide variants previously described by us and others[39, 40]: the slowly aggregating $A\beta_{40}$ peptide (target) and the highly aggregation-prone $A\beta_{42}$arctic variant (seed). To mimic its physiological neuronal release, all $A\beta$ variants are secreted into the extracellular space using signal peptide fusion constructs[39]. Using the driver lines described above (Fig. 1), expression of highly toxic $A\beta_{42}$arctic was only induced in restricted neuronal clusters using either the *R9D03*-Gal4 or *GMR14B06*-Gal4 driver, whereas $A\beta_{40}$ was expressed pan-neuronally using the *nSyb*-LexA driver. To investigate the effect of co-expressing both $A\beta$ variants, we analyzed the levels of soluble and insoluble $A\beta$ in brain homogenates using western blot analysis (Fig. 2a, b;

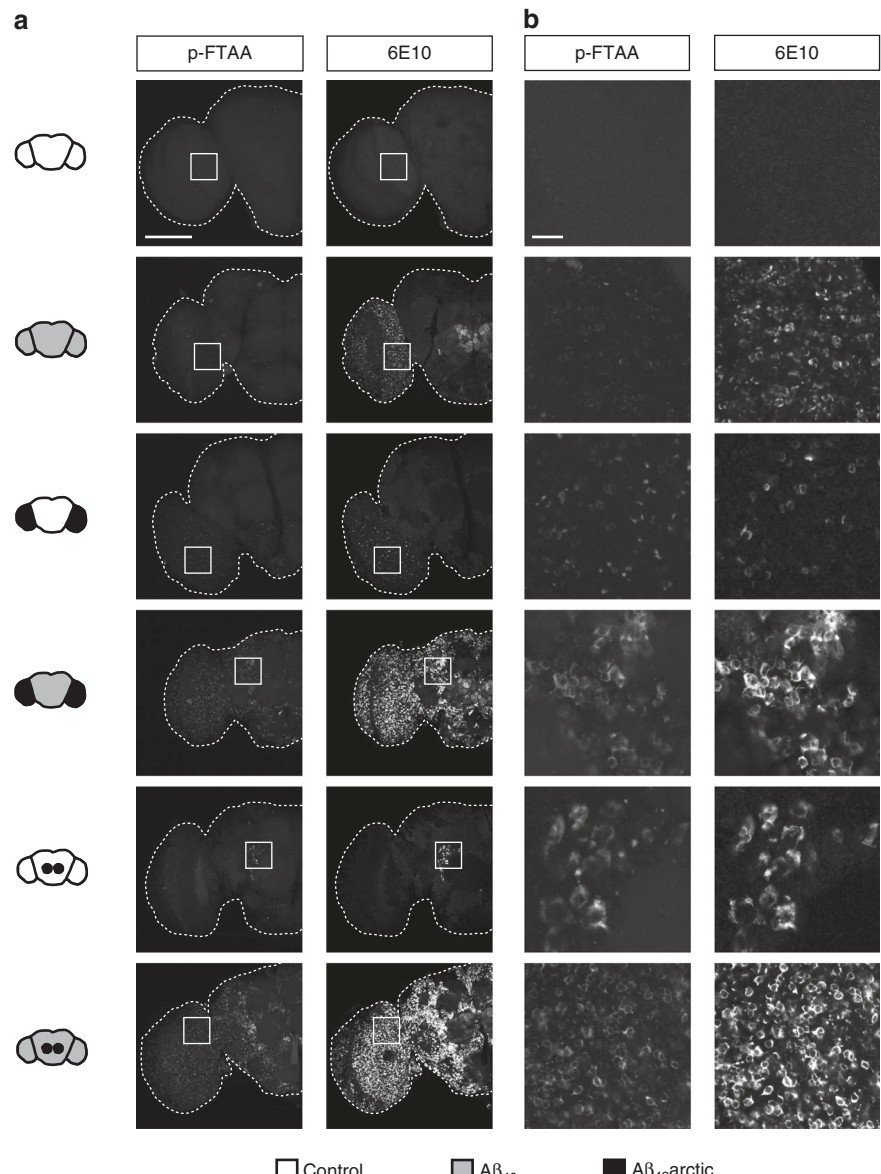

**Fig. 3** Amyloid aggregates are formed throughout the brain. **a** *Drosophila* brains were dissected at day 21, stained with the monoclonal Aβ antibody 6E10 and the amyloid-specific polymer probe p-FTAA and imaged using confocal microscopy. Control represents driver line only. *Scale bar*, 100 μm. **b** Zoom of the region indicated in **a** (*white square*). *Scale bar*, 10 μm. In total eight brains were analyzed per genotype (penetrance of the phenotype = 100%)

Supplementary Fig. 1a, b). We did not observe significant accumulation of insoluble Aβ40 when expressed pan-neuronally, albeit its high abundance as monomeric, soluble peptide at an early time point. Over time the level of soluble Aβ40 decreased (Supplementary Fig. 1), which likely results from the efficient clearance of the peptide and a reduced activity of the LexA driver at later age. The restricted expression of Aβ42arctic in a subset of neurons in the optic lobes (using *R9D03*-Gal4) resulted in the formation of small amounts of insoluble Aβ seeds (Fig. 2a). Consistent with its restricted expression pattern, insoluble Aβ42arctic was below the detection limit when induced by the *GMR14B06*-Gal4 driver only (Fig. 2b), even in 21-day-old flies. Next, we performed the analysis in flies expressing both Aβ variants. Intriguingly, introducing Aβ42arctic seeds into flies expressing Aβ40 pan-neuronally resulted in an increase in insoluble Aβ over time. This time-dependent Aβ deposition is observed in both dual expression systems, despite the very low amount of initial seeding material. These findings strongly

suggest a seeding-based aggregation reaction, as accelerated Aβ40 deposition was only observed when Aβ42arctic seeds were expressed simultaneously.

To examine the localization of these deposits in the fly brain, we further assessed our models via immunohistochemistry. We dissected *Drosophila* brains at day 21 post eclosion and visualized Aβ using the 6E10 monoclonal Aβ antibody[41]. Pan-neuronal expression of Aβ40 gave rise to a weak 6E10 signal throughout the brain (Fig. 2c) in accordance with an overall low amount of total Aβ40 at 21 days post eclosion (Supplementary Fig. 1). In flies expressing only the Aβ42arctic variant, we observed a confined Aβ signal restricted to the optic lobes (*R9D03*-Gal4) or the central brain (*GMR14B06*-Gal4) (Fig. 2c). In line with our biochemical data, increased Aβ accumulation was observable throughout the brain of flies expressing the two Aβ variants simultaneously. This finding suggests that insoluble Aβ detected by western blot analysis is not restricted to the site of seed expression, but can be found throughout a broad cerebral area.

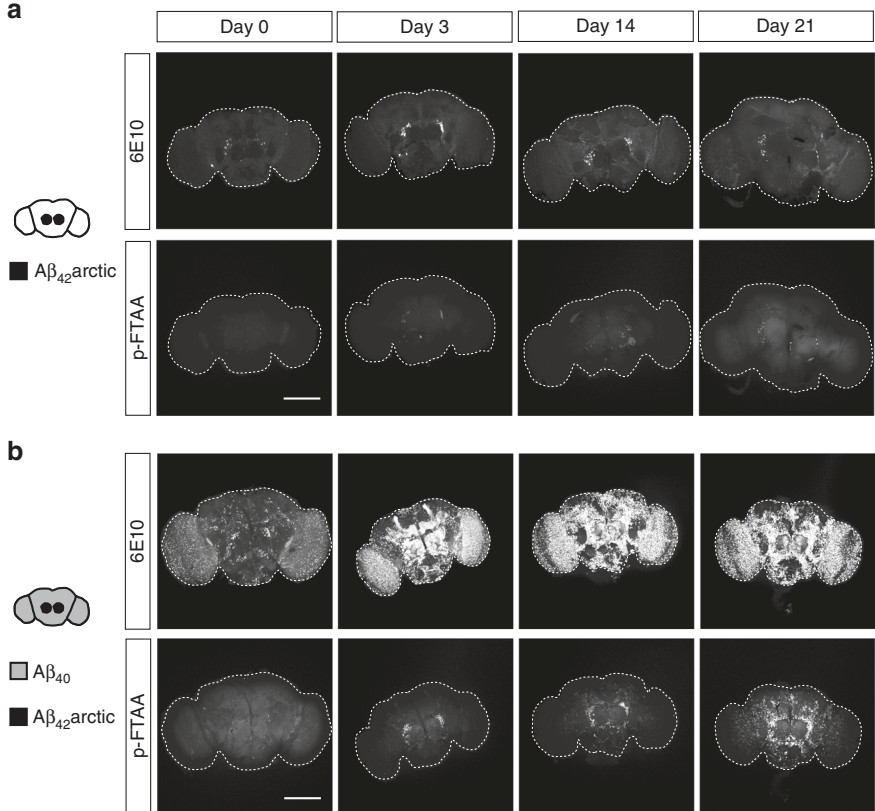

**Fig. 4** Aβ deposits are formed throughout the brain starting at the site of seed expression. **a**, **b** Confocal images of brains from **a** flies expressing only Aβ$_{42}$arctic in a subset of cells in the central brain (driven by *GMR14B06*-Gal4), and **b** flies expressing Aβ$_{40}$ (as target) and Aβ$_{42}$arctic (as seed) simultaneously under control of the double-driver *nSyb*-LexA::*GMR14B06*-Gal4. *Drosophila* brains were dissected at the indicated time points (days post eclosion) and stained with the 6E10 antibody recognizing total Aβ and the amyloid-specific polymer probe p-FTAA. In total, nine brains were analyzed per genotype and time point and the penetrance of the phenotype was 100%. *Scale bars*, 150 μm

Aβ has been suggested to accumulate extra- and intracellularly in vivo, however it remains unclear to what level Aβ is taken up by neurons and how the different Aβ pools contribute to seeding processes[42]. Analyzing the localization of Aβ quantitatively in our *Drosophila* models has proven difficult, given the small size of neuronal cell bodies. However, Aβ staining using 6E10 (Fig. 2c) expectedly showed a different pattern than intracellularly expressed mCherry (Fig. 1). To get an insight into how secretion contributes to the cellular localization of Aβ, we generated a *Drosophila* line expressing Aβ$_{42}$ missing the secretion peptide (Aβ$_{42}$NSP). Immunohistochemical analysis of fly brains revealed that Aβ accumulation only occurs when Aβ$_{42}$ carries the signal peptide, that is, when it is targeted for secretion (Supplementary Fig. 2a), whereas Aβ$_{42}$NSP cannot be detected (Supplementary Fig. 2b), suggesting its efficient degradation inside cells. These data support the notion that extracellular Aβ is required for efficient seeding and that Aβ accumulations masks the outside of neurons in our *Drosophila* models.

Notably, a direct seeding mechanism does not only imply that more Aβ is deposited, but also that aggregation of the normally soluble Aβ$_{40}$ is induced. To address this aspect, we performed high sensitivity Electrochemiluminescence (ECL) detection assays allowing the variant-specific measurement of Aβ levels. Consistently, the amount of insoluble Aβ$_{40}$ was significantly increased (up to 20-fold) in both fly models expressing seed and target simultaneously, in comparison to flies expressing Aβ$_{40}$ alone (Fig. 2d). This observation provides strong evidence for the incorporation of normally soluble Aβ$_{40}$ into insoluble deposits in the presence of Aβ$_{42}$ seeds, supporting the relevance for a

templated conversion mechanism in a physiologically relevant in vivo setting. To further evaluate whether more indirect proteostasis mechanism or cellular stress signaling events contribute to the observed seeding effect, we also generated transgenic fly lines expressing the aggregation-prone Huntingtin (Htt) protein, containing an expanded polyQ tract (HttQ72), as seeds in combination with the Aβ$_{40}$ target peptide. Subsequently, we analyzed levels of insoluble Aβ$_{40}$ in this genetic setup in the presence of either Aβ$_{42}$arctic or HttQ72 seeds. Importantly, HttQ72 seeds did not lead to an increase in the amount of total Aβ (Supplementary Fig. 3a) nor in the level of insoluble Aβ$_{40}$ (Supplementary Fig. 3b). In contrast, we observed an increase in the amount of insoluble Aβ when using Aβ$_{42}$arctic seeds in combination with the target peptide. This finding suggests that the seeding effect observed here is not simply caused by general alterations in neuronal proteostasis, but that it is specifically induced by a templated aggregation process.

**Amyloid deposition propagates throughout the brain.** To further examine the nature of the deposits, we stained brains of 21-day-old flies with the 6E10 Aβ antibody and additionally with the amyloid-specific p-FTAA polymer probe (Fig. 3). This probe belongs to a group of luminescence-conjugated oligothiophenes, where binding to protein aggregates induces a structural restriction of the backbone[43, 44], resulting in a specific emission spectrum of the probe. Previous studies showed that p-FTAA reliably labels Aβ aggregates in the fly brain with high sensitivity and a good signal-to-noise ratio[43], demonstrating its suitability

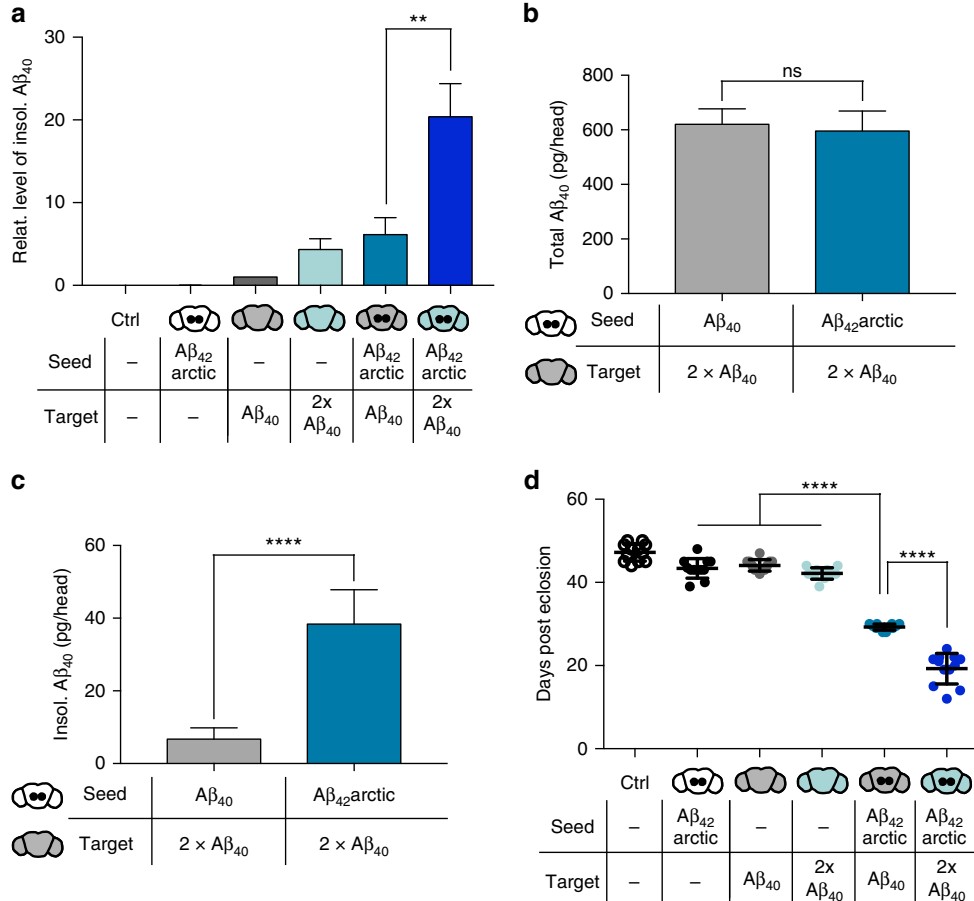

**Fig. 5** In vivo seeding of Aβ aggregation is dose-dependent and causes neurotoxicity. **a** ECL measurement of insoluble $A\beta_{40}$ in 21-day-old flies with the indicated genotypes. Expression of $A\beta_{42}$arctic seeds was driven using the *GMR14B06*-Gal4 driver, whereas $A\beta_{40}$ was expressed pan-neuronally with the *nSyb*-LexA driver. Level are normalized to $A\beta_{40}$ only (error bars, s.e.m.; $n = 3$ independent biological replicates, one-way ANOVA (Tukey's multiple comparison test), **$P = 0.0022$). For the matter of clearness not all significances are indicated. **b**, **c** ECL measurement of total (**b**) or insoluble (**c**) $A\beta_{40}$ levels in 21-day-old flies expressing two copies of the $A\beta_{40}$ target peptide in addition to either $A\beta_{40}$ "control seeds" or $A\beta_{42}$arctic seeds. Seed expression was driven using *GMR14B06*-Gal4. Total $A\beta_{40}$ represents the sum of soluble and insoluble $A\beta_{40}$ levels (error bars, s.d., $n = 3$ independent biological replicates, unpaired Student's *t*-test, $^{ns}P = 0.527$, ****$P = 0.0001$). **d** Median survival of flies analyzed in **a** (error bars, s.d., $n = 100$ independent biological replicates, one-way ANOVA (Tukey's multiple comparisons test), ****$P = 0.0001$). For the matter of clearness not all significances are indicated

for application to *Drosophila* models. We observed that structures strongly labeled with 6E10 were also positive for p-FTAA, demonstrating the amyloid character of these deposits (Fig. 3a, b). In flies expressing only $A\beta_{42}$arctic in small neuronal clusters, these amyloid aggregates were constricted to the target region in either the optic lobes or the central brain. Notably, we observed a strong increase in Aβ staining as well as p-FTAA positive material throughout the whole brain in flies simultaneously expressing seed and target. Clear deposition of amyloid material can even be observed in distal regions, up to several hundred micrometers from the initial site of seed generation, demonstrating profound propagation of amyloid pathology to distant areas of the brain.

To get more detailed insights into the early stages preceding such a severe Aβ deposition, we analyzed the distribution of Aβ aggregates at earlier time points (Fig. 4). Flies expressing $A\beta_{42}$arctic in a subset of neurons in the central brain showed Aβ staining restricted to this expression region (Fig. 4a). The p-FTAA signal is weak in these flies, confirming the small amount of available seeds. Interestingly, in flies expressing seed and target simultaneously, amyloid aggregates first appear in the area of seed expression, but can also be found in more distal brain areas at later time points (Fig. 4b). Here, a clear increase in overall 6E10

immunoreactivity throughout different brain regions could be observed over time which was reflected by the increased distribution of pFTAA-positive amyloid material, confirming our biochemical data. These results are consistent with a specific, locally restricted initiation of aggregation, in regions where $A\beta_{42}$arctic seeds are generated. Originating from this area, amyloid aggregates appear throughout the brain over time, pointing to sequential seeding mechanisms. This suggests that a small amount of aggregated material is sufficient to initiate the aggregation of the normally soluble peptide in a confined cerebral area, resulting in more aggregated and seeding-competent material that can further drive the seeding cascade in associated brain regions.

**Secondary nucleation drives the seeded deposition of $A\beta_{40}$.** Secondary nucleation mechanisms have been suggested as the major driving force in progressive protein aggregation in vitro and in vivo[22, 23], where the rate-limiting factor for aggregation becomes the availability of soluble peptide monomers[22, 23, 25]. The severity of protein seeding, therefore, is tightly dependent on the amounts of the introduced seed as well as the amount of available target[16, 32, 45, 46]. In addition, intriguing evidence

suggests that the amount of the target peptide is crucial for the formation of toxic protein assemblies, for example of $A\beta$[23] and the prion protein[45]. As an added insight into this mechanism in vivo, we tested this hypothesis in our *Drosophila* models by expressing two copies of the target peptide pan-neuronally. To analyze the influence of this augmented target expression on $A\beta$ aggregation in our seeding model, we performed high sensitivity ECL measurements (Fig. 5a). Strikingly, expression of an additional copy of the target peptide in combination with $A\beta_{42}$arctic seeds, resulted in an approximately threefold increase in the amount of insoluble $A\beta_{40}$, compared to flies expressing only one copy of $A\beta_{40}$ (Fig. 5a). Hence, our results demonstrate that increased pan-neuronal expression of a non-aggregation-prone $A\beta$ variant is sufficient to induce a significant aggravation of the seeding phenotype resulting in augmented deposition of the otherwise soluble $A\beta_{40}$. Notably, this effect is not caused by a mere increase in the amount of total $A\beta_{40}$, but is specifically induced by $A\beta_{42}$arctic seeds (Fig. 5b, c). Here, expression of $A\beta_{40}$ "control seeds" and $A\beta_{42}$arctic seeds in the central brain in addition to two copies of the target peptide resulted in similar levels of total $A\beta_{40}$ (Fig. 5b) but only the presence of $A\beta_{42}$arctic seeds led to a significant increase in the amount of insoluble $A\beta_{40}$ (Fig. 5c). The finding that the seeding effect depends strongly on the amount of available target peptide supports the relevance for secondary nucleation processes in the progression of $A\beta$ aggregation.

Of further note, detailed analysis of plaque composition in AD patients revealed the accumulation of different $A\beta$ species within these deposits[47, 48]. In accordance, we observed an augmented deposition of $A\beta_{42}$ in flies expressing seed and target (Supplementary Fig. 4a), although this $A\beta$ variant is only expressed at low levels in this system and hardly accumulates in the insoluble fraction when expressed exclusively (Fig. 2a, b; Supplementary Fig. 4a). Overall, these findings suggest that the induced seeding phenotype in our fly model leads to the stabilization not only of $A\beta_{40}$ but also of $A\beta_{42}$arctic within the aggregates, further increasing the overall $A\beta$ burden. Analysis of the level of soluble $A\beta$ shows that $A\beta_{40}$ was present at much higher levels than the $A\beta_{42}$arctic variant (Supplementary Fig. 4b), demonstrating that the original expression setup with low amounts of peptide seeds and an abundant target peptide persisted.

**Seeded $A\beta$ deposition reduces survival.** A crucial question for the relevance of $A\beta$ aggregate propagation in AD, is the impact of seeded amyloid deposition on neurodegeneration[26, 49–51]. While monitoring neurotoxicity has proven challenging in established animal models for $A\beta$ seeding due to the complexity of the model organisms[15, 16, 29–32], our approach allows investigating $A\beta$-induced neurotoxicity on a short time scale. Here, we assessed neurotoxic effects by performing survival assays (Fig. 5d), a robust and broadly validated readout for protein aggregation-mediated neurotoxicity[39, 40, 52, 53]. Consistent with previous data[39, 40], and in line with the absence of insoluble amyloid material (Fig. 2), the pan-neuronal expression of $A\beta_{40}$ did not have a strong effect on fly survival (Fig. 5d). Despite its severe toxic potential, when expressed pan-neuronally[40], expression of $A\beta_{42}$arctic in restricted neuronal clusters did not impact fly survival, as only a small cerebral area was affected and the overall amount of aggregated $A\beta_{42}$arctic is minor (Figs. 2a, b). We observed significantly reduced survival of flies expressing both seed and target (median survival $29 \pm 0.7$ days) compared to flies expressing only one of the $A\beta$ variants (median survival $43 \pm 2.4$ days and $44 \pm 1.4$ days $A\beta_{42}$arctic and $A\beta_{40}$ for respectively; Fig. 5d). Previous studies mainly assessed toxicity of seeding

processes by analyzing local brain pathology, such as dystrophic neurites in the vicinity of $A\beta$ deposits[54, 55]. Our finding now demonstrates for the first time that in vivo $A\beta$ seeding mechanisms are linked to severe reduction of lifespan.

To further confirm the correlation between decreased survival and seed-induced $A\beta$ accumulation, we analyzed the survival rate in the double target system (Fig. 5d). Expression of two copies of the target peptide in the absence of seeds did not lead to a reduced survival (median survival $42 \pm 1.4$ days). Strikingly, expression of two copies of $A\beta_{40}$ in combination with $A\beta_{42}$arctic seeds resulted in an aggravated survival phenotype (median survival $20.5 \pm 3.7$ days). Thus, by increasing the amount of the otherwise innocuous $A\beta_{40}$ peptide in the presence of the seed, toxicity is further enhanced, supporting the finding that the amount of the soluble target peptide plays a decisive role in the formation of toxic protein species[23, 45].

The $A\beta_{42}$arctic variant has been described only in a few familial cases of AD, whereas the non-mutation carrying $A\beta_{42}$ represents the major disease-related variant in sporadic AD. To test whether our setup is also sensitive to study seeding effects of this less aggregation-prone and less toxic $A\beta$ species, we locally introduced $A\beta_{42}$ seeds in the central brain in the background of one or two copies of the target peptide (Fig. 6). ECL measurements revealed that also $A\beta_{42}$ seeds were potent to induce increased deposition of $A\beta_{40}$ (Fig. 6a), which increased more than 20-fold compared to controls, thereby exceeding the seeding effect induced by $A\beta_{42}$arctic seeds. Possible reasons for this effect could be differential structural properties or temporal abundance of seeds generated from these different $A\beta_{42}$ variants. Importantly, $A\beta_{42}$-induced deposition of $A\beta_{40}$ again correlates with neurotoxicity, indirectly measured by survival assays, in a dose-dependent manner (Fig. 6b). The survival of flies expressing $A\beta_{42}$ seeds in addition to one copy of the target peptide was severely reduced (median survival $40.7 \pm 0.7$ days) in comparison to flies expressing only the target (median survival $53.1 \pm 2.4$ days). The expression of two copies of the target peptide further aggravated neurotoxicity (median survival $31.2 \pm 1.4$ days). The survival phenotype in the presence of $A\beta_{42}$arctic was significantly more prominent (median survival $34.1 \pm 1.7$ days with one copy of the target peptide or $28.6 \pm 1.5$ days with two copies of $A\beta_{40}$) than the one caused in the presence of $A\beta_{42}$ seeds, which again may relate to distinct characteristics of the formed $A\beta$ aggregates. To explore a complementary readout to confirm the neurotoxicity described by our survival analysis, we performed an automated locomotion analysis[40] for this set of *Drosophila* lines. Here, we replicated the strong impact of $A\beta_{42}$ seeds, with fly locomotion being severely impaired already at day 14 of analysis (Fig. 6c). This phenotype was again aggravated in the presence of two copies of the target peptide.

Taken together, our findings demonstrate a link between the seed-induced acceleration of $A\beta$ pathology and associated neurotoxicity, as measured indirectly by decreased fly locomotion and severely reduced life span of flies. To our knowledge, this is the first study directly linking $A\beta$ seeding processes to reduced lifespan and behavioral deficits in a closed biological system, giving further emphasis to the potential of these novel *Drosophila* models for studying the molecular basis of $A\beta$ seeding mechanisms in vivo.

**Discussion**

The deposition of $A\beta$ aggregates in senile plaques, follows a stereotypical progression throughout the AD brain over time[7–9, 56]. Until now, the exact mechanisms underlying this spreading of pathology are not clarified, but a process termed seeded nucleation has been suggested to play a major role[9, 13, 14].

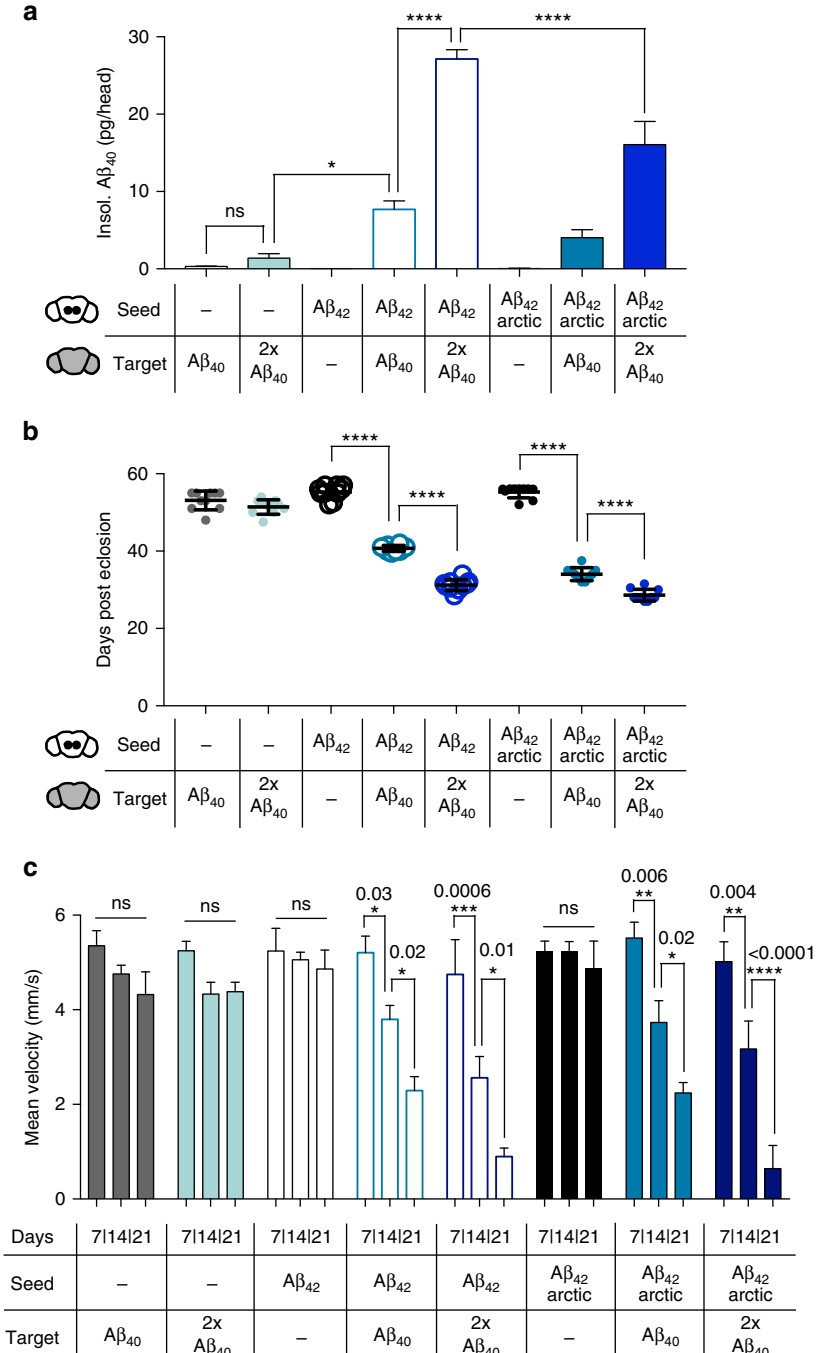

**Fig. 6** In vivo seeding models comparing the seeding effect of $A\beta_{42}$ and the mutated $A\beta_{42}$arctic peptide. **a** Quantification of insoluble (insol.) $A\beta_{40}$ in 21-day-old flies with the indicated genotypes using ECL (error bars, s.e.m, $n = 4$ independent biological replicates, one-way ANOVA (Tukey's multiple comparisons test, $^{ns}P = 0.9988$, $*P = 0.0158$, $****P < 0.0001$). **b** Median survival of flies analyzed in **a** (error bars, s.d.; $n = 82$ independent biological replicates, one-way ANOVA (Tukey's multiple comparisons test), $****P < 0.0001$). **c** Mean velocity of flies analyzed in **a**, **b** at the indicated time points (days post eclosion) using the automated iFly locomotion setup (error bars, s.e.m., $n = 20$, two-way ANOVA (Tukey's multiple comparisons test), the numbers above the asterisks indicate the P-value). For clearance not all significances are shown **a**–**c**

According to this concept, minor levels of aggregated Aβ act as a template to seed misfolding and aggregation of the otherwise soluble cognate peptide, resulting in a dramatic increase in the number of Aβ deposits in the brain over time[17]. Although several animal models support this idea[15, 16, 29–32], the exact processes underlying this phenomenon remain unknown, as detailed mechanistic studies are impeded due to the complexity of the model organisms. Here, we introduce in vivo Drosophila models

to study Aβ seeding mechanisms and subsequent neurotoxicity effects on a short time scale. In contrast to published animal seeding models, which are based on the injection of seeding-competent material into the host organism, the Drosophila models presented here are fully genetically encoded, allowing the study of Aβ seeding processes in an intact neuronal system, without the requirement for an invasive inoculation with the seed. Local production of minor amounts of aggregation-

prone peptide variants might be caused by mosaic mutations occurring in AD patients' brains[27, 28] and by local changes in protein homeostasis in confined neuronal subpopulations. Such a heterogeneous expression pattern is mimicked in our fly models (Fig. 1), given that an aggregation-prone peptide species, here variants of the 42 amino acid long Aβ42 peptide, are expressed in a restricted number of neurons, whereas Aβ40 is expressed pan-neuronally. The Aβ42arctic variant was initially chosen as a proof-of-concept, due to its implication in familial cases of AD and its increased aggregation propensity[57, 58]. However, this concept is replicated for the non-mutated Aβ42 peptide, which represents the major deposited species in sporadic late onset AD. Aβ40 was chosen as the target peptide, since it is the most abundant Aβ species in the human brain and it is slowly aggregating as well as non-toxic when expressed pan-neuronally in flies.

In contrast to previous co-expression studies[53], our models describe specific seeding models, where very low amounts of seed and comparably high levels of the target peptide mimic a classic seeding environment. Importantly, we observed a time-dependent increase of insoluble Aβ40 when introducing the locally restricted expression of Aβ42 seeds simultaneously in the fly brain (Fig. 2a, b). This seeding effect was observed in both expression systems, regardless of expressing the seed in a neuronal cluster in the optic lobes or the central brain. Interestingly, amyloid deposition was not locally restricted to the site of initial seed expression, but was distributed throughout the brain (Fig. 2c). Similar effects were observed in rodent models, exhibiting an accelerated formation of Aβ pathology in inoculated animals, with lesions concentrated at the injection site but also reaching more distal regions[15, 16, 32]. These shared features confirm our Drosophila setup as suitable seeding model systems, allowing the study of complex phenotypes in a versatile and genetically powerful setting.

Templated misfolding and aggregation of a normally soluble protein variant is a basic element of the seeding concept[22, 59]. Consistent with this idea, we robustly observed insoluble Aβ40 in flies expressing seed and target simultaneously (Fig. 2d), clearly indicating that minor amounts of seeds are sufficient to induce the deposition of the otherwise soluble Aβ40. Hence, the here presented Drosophila models recapitulate the basic Aβ seeding principle from in vitro studies to a complex in vivo setting. In addition, we observed significant amounts of insoluble Aβ42 in these double transgenic flies, despite its low expression level and its minor deposition when expressed exclusively. In AD patients, Aβ42 was identified as a major component of senile plaques[48, 60–62], albeit being generated to much lower levels compared to Aβ40[63, 64]. Thus, the overall stabilization of Aβ42 in insoluble protein accumulations appears to be a common mechanism in the development of Aβ pathology, which has also been observed in rodent seeding models[16, 31]. Analyzing the soluble protein fraction in this respect revealed up to 17-fold higher levels of Aβ40 compared to Aβ42, demonstrating the persistence of the seeding environment in this model, characterized by low amounts of seeds in the presence of large quantities of the target peptide.

Development of amyloid aggregates are a hallmark of AD pathology[9] and post-mortem studies, revealed a characteristic spreading pattern of Aβ pathology throughout the brain of AD patients[7, 8]. The progressive formation of amyloid deposits could be visualized in our seeding models by staining with an amyloid-specific polymer probe (Fig. 3). Remarkably, we observed a time-dependent expansion of Aβ pathology throughout the brain starting at the site of initial seed expression (Fig. 4). While the detailed mechanism remains to be determined, we envision a consecutive seeding process as possible underlying cause, in which initially only a small portion of soluble protein

is driven to aggregation[22]. Progressive fibril fragmentation might further give rise to an increase in seeds available to act as templates for seeded conversion of the monomeric peptide in a process termed secondary nucleation[22, 25]. Recent data implicate secondary nucleation mechanisms as the driving force in certain seeding processes, with the amount of available target protein being the rate-limiting factor[22, 23, 25]. Along these lines, previous studies have shown that increasing the amount of the target peptide enhances protein deposition and formation of toxic peptide species[45, 65]. Confirming this concept in our Drosophila model, we observed that enhancing the amount of the target peptide leads to a further increase in the deposition of Aβ (Fig. 5a). This finding is in line with theoretical predictions concerning the kinetics of fibril assembly, which include the concentration of available monomers as a critical variable[22]. Hence, our study suggests that the general concept of Aβ fibril formation following secondary nucleation pathways[25] is applicable to biological in vivo settings.

One of the challenging questions in AD research is the role of different disease-related Aβ assemblies in toxicity[49–51]. The Drosophila models described in this work allow the detailed mechanistic study of seeded Aβ deposition and resulting neurotoxicity effects. For the first time, we could demonstrate a direct link between seed-induced acceleration of Aβ deposition and reduced survival of flies expressing seed and target simultaneously (Fig. 5d). This effect was dose-dependent as flies expressing higher levels of the target peptide, resulted in elevated Aβ deposition and an aggravated survival phenotype. Our results, therefore, mechanistically connect aggregate deposition and decreased lifespan, although the underlying aggregation species remains to be determined in future studies. As suggested for other systems, the species relevant for seeding processes might not be the toxic culprit[45]. Future studies are required to reveal to what extent these findings, concerning Aβ species determined neurotoxicity in Drosophila, can be transferred to mammalian systems and AD patients.

In conclusion, we introduce novel non-invasive Drosophila models, where both seed and target are fully genetically encoded, to reveal mechanisms involved in seed-induced templated protein misfolding in an intact neuronal system. Our results provide strong evidence for the common applicability of the seeding concept, further confirming the role of secondary nucleation processes as basic mechanistic drivers behind this phenomenon. Importantly, we support the relevance for these seeding processes in disease progression, by showing a direct link between seed-induced Aβ deposition and reduction in survival, an indirect measure for neurotoxicity. These results highlight the value of fly models for studying the mechanisms leading to neurodegeneration caused by Aβ. Notably, these models are clearly amenable to study the role of protein aggregation and propagation of pathology for a variety of other proteins associated with neurodegeneration, and will provide crucial mechanistic insights into the etiology of this devastating class of diseases.

## Methods

**Transgenic Drosophila melanogaster lines.** The human Aβ40, Aβ42, and Aβ42arctic (Aβ42 E22G) sequences were cloned downstream of a signal peptide derived from the Drosophila necrotic gene[66], allowing efficient secretion of Aβ upon expression. The pJFRC7 vector[67] (Addgene, cat. no. 26220) was used for expression under control of the Gal4/UAS expression system, and the pJFRC19 vector[67] (Addgene, cat. no. 26224) for expression under control of the complementary LexA/LexAop system. To analyze how secretion influences the deposition of Aβ we furthermore generated constructs that contain Aβ42 without the secretion peptide (NSP). Transgenic flies were generated by phiC31 integrase-mediated transgenesis[68] using attP landing sites 25C6 (second chromosome) and 68A4 (third chromosome). Expression was driven pan-neuronally using the nSyb-LexA driver line, or in selected neuronal clusters using R9D03-Gal4[37] or GMR14B06-Gal4[38] (Bloomington). To generate fly lines expressing Aβ42arctic or

Huntingtin (HttQ72) seeds in the central brain in addition to the pan-neuronally expressed target peptide (Supplementary Fig. 3), we crossed the previously generated triple transgenic flies w;LexAop-Aβ$_{40}$;nSyb-LexA::GMR14B06-Gal4 to w; UAS-Aβ$_{42}$arctic or w;UAS-HttQ72, respectively. Flies were raised on standard cornmeal and molasses medium. Crosses were kept for 3 days at 25 °C and then shifted to 29 °C (60% rH). In variation from this, flies were first kept at 25 °C for 3 days and subsequently transferred to 18 °C for the earliest time point (day 0) in the time-course experiment (Fig. 4) in order to reduce expression to a minimum. The F1 was collected in 24 h windows and mated females were kept at 29 °C before freezing at -80 °C at indicated time points.

**Differential extraction of Aβ from fly head homogenates**. Flies were decapitated and heads were lysed in Buffer 1 (50 mM Tris (pH 7.5), 2 mM sodium orthovanadate, 50 mM sodium fluoride, 50 mM β-Glycerophosphate disodium salt hydrate, 1× phosphatase inhibitor (Roche), 1× protease inhibitor (Santa Cruz Biotechnology), 150 mM sodium chloride, 2 mM magnesium chloride, 1% (w/v) N-lauroylsarcosine, 1% (v/v) Triton X-100 and 1% (w/v) sodium dodecyl sulfate (SDS)) using a Minilys personal homogenizer (Peqlab). For western blot analysis, we started with 25 heads and for ECL measurements we used up to 50 heads per sample. After homogenization, the samples were sonicated for 15 min and further incubated on ice for 15 min. Tissue debris was removed by centrifugation at 3800 × g for 5 min at 4 °C. Overall protein concentrations were determined using Lowry quantification (DC Protein Assay, BioRad) and adjusted accordingly. Total protein (115 μg) was used for western blot analysis and between 160 and 260 μg for ECL quantification. Samples were further incubated in the presence of 1% (v/v) β-Mercaptoethanol for 1 h on ice, before differential centrifugation with 21,000 × g at 4 °C for 1 h to yield soluble and insoluble protein fractions.

**Western blot analysis**. The soluble protein fraction was directly incubated for 5 min at 95 °C in 1× Lämmli (60 mM Tris-HCl (pH 6.8), 2% SDS, 10% Glycerol, 5% β-Mercaptoethanol, 0.01% bromophenol blue). The insoluble pellet fraction was resuspended in 400 μl of Buffer 1 to remove contaminations, followed by sonication for 15 min. Afterwards, the samples were centrifuged at 21,000 × g for 30 min at 4 °C. This washing step was repeated two times. The resulting pellet was resolubilized using 100% DMSO and incubated for 1 h at 25 °C. Subsequently, the DMSO was diluted (2.4-fold) by adding 1× Lämmli in Buffer 1, followed by incubation at 95 °C for 5 min. Samples were analyzed using NuPAGE Novex 4–12% Bis–Tris gels and NuPAGE MES SDS Running Buffer (Life Technologies). The Spectra Multicolor Low Range Protein Ladder (Thermo Fisher Scientific) was used as size marker (M). The semi-dry transfer onto Amersham Protran 0.1 μm Nitrocellulose membrane (GE Healthcare) was performed using a Trans-Blot Turbo Transfer System (BioRad). After transfer, membranes were shortly boiled in phosphate buffered saline (PBS) for antigen retrieval and blocked in 5% (w/v) milk powder in PBS + 0.1% Tween-20 for 1 h at room temperature (RT). The membranes were further incubated in primary antibody against Aβ (6E10, 1:600, Covance) or against Glyoxalase 1 (Glo1, 1:1500, Santa Cruz Biotechnology Inc.) as a loading control over night at 4 °C. After washing, incubation in secondary antibody (goat α-mouse IgG-HRP 1:2000, goat α-rabbit IgG-HRP 1:3000, Invitrogen) was performed for 2 h at RT. Subsequently, the membranes were incubated with SuperSignal West pico or femto Chemiluminescent Substrate before visualization using a C-DiGit Blot Scanner (LI-COR). Quantification of western blots as well as contrast and brightness adjustments of the images were performed using the Image Studio Lite Software (LI-COR). Aβ levels were normalized to Glo1 to exclude effects of unequal loading.

**Electrochemiluminescence detection assay**. Electrochemiluminescent (ECL) detection of Aβ was carried out using the V-PLEX Aβ Peptide Panel 1 (6E10) Kit (Meso Scale Discovery) according to the manufacturer's manual. The insoluble pellet fraction was resuspended in 4 M Guanidin hydrochloride (GdnHCl, Carl Roth) in 50 mM Tris pH 7.4, 1 mM EDTA, 1× protease inhibitor (Santa Cruz Biotechnology). The soluble protein fraction was diluted 1:1 in the same buffer as the insoluble fraction, however containing 8 M GdnHCl. Next, the samples were incubated at 25 °C for 1 h (shaking) and subsequently diluted 1:1 in Diluent 35 (V-PLEX Aβ Peptide Panel 1 (6E10) Kit, Meso Scale Discovery) to reduce the GdnHCl concentration to 2 M. Then, samples were incubated at 25 °C for 30 min, sonicated for 5 min and stored at −20 °C. After thawing, the samples were further diluted in Diluent 35 to reduce GdnHCl to maximum 250 mM. The ECL signal was measured using the MESO QuickPlex SQ 120.

**Locomotion assay**. Fly locomotion was assessed using the iFly setup described previously[40]. Climbing trajectories of 10 flies per measuring tube were recorded on day 7, 14, and 21 over a 45 s period and analyzed using an in house developed software[40].

**Dissection and whole brain 6E10 and p-FTAA staining**. Adult fly brains were dissected according to Wu and Luo[69] with the following changes: flies were dissected in PBS and kept in PBST (PBS + 0.5% (v/v) Triton-X 100) until fixation. Brains were fixed by incubation in freshly prepared 3.7% (w/v) Formaldehyde solution (Sigma) in PBST for 30 min, followed by two 5 min and two 15 min

washes with PBST. Brains expressing green fluorescent protein (GFP) and mCherry were directly mounted in Vectashield (Vector Laboratories). For immunohistochemistry, the brains were dehydrated in Methanol (successive steps in 30, 50, 70, and 100% Methanol in PBST, 30 min at 4 °C each) and stored at −20 °C. Before further staining, the brains were rehydrated by following the Methanol steps in reverse order. Subsequently, the brains were incubated in 5% (v/v) FBS in PBST (blocking solution) for 1 h at RT before addition of the primary antibody α-6E10 (1:1500, Covance) and an incubation of 48 h at 4 °C. To remove the primary antibody the brains were washed two times for 5 min in PBST followed by a washing step over night at 4 °C and a washing step for 1 h at RT. Afterwards, the brains were incubated in Goat anti-Mouse IgG (H + L) Secondary Antibody, Alexa Fluor 568 conjugate (1:1000, Thermo Fisher Scientific). Staining with p-FTAA has been described previously[43, 44]. We adapted the protocol with the following changes: 3 μM p-FTAA were added to the dilution of the secondary antibody and the brains were incubated for approximately 30 h at 4 °C. Finally, the brains were washed as described above (after primary antibody incubation), mounted in Vectashield and imaged using a Zeiss LSM 780 Laser Scanning confocal microscope with the Software ZEN 2010 B SP1 or ZEN 2.1. Images were processed using Fiji[70] and Adobe Photoshop CS3. Adjustments of brightness and contrast were applied equally across each image and were also applied equally to controls.

**Statistical analysis**. Statistical analysis was performed using the GraphPad Prism Software. Error bars indicate either standard deviation (s.d.) or standard error of the mean (s.e.m.). Only positive error bars are shown for simplification, however, they also symmetrically go in the negative direction. The exact sample numbers and P-values as well as the applied statistical tests are clarified in the figure legends (not significant $^{ns}P > 0.05$; $*P ≤ 0.05$; $**P ≤ 0.01$; $***P ≤ 0.001$; $****P ≤ 0.0001$).

**Data availability**. The data sets generated and analyzed during this study are available from the corresponding author on reasonable request.

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

## Acknowledgements

This work was supported by the Chica and Heinz Schaller Foundation and an Annemarie Poustka Fellowship (by the Helmholtz International Graduate School for Cancer Research, DKFZ) to R.F.S. We thank the University of Cambridge Department of Genetics Fly Facility, the Bloomington Drosophila Stock Center, B. Pfeiffer (Janelia Research Campus) and P. Hammarström (Linköping University) for reagents and T. Johnson for providing the MESO QuickPlex SQ 120 for ECL measurement.

## Author contributions

Experimental work was performed by R.F.S.; T.R.J. designed and supervised the study; R.F.S. and T.R.J. wrote the manuscript.

## Additional information

**Competing interests:** The authors declare no competing financial interests

