## [Peer Review File · Nature Communications]

Reviewers' comments:

Reviewer #1 (Remarks to the Author):

This is an interesting report which provides proof of concept data for a drosophila model system for studying the relationship between amyloid-beta seeding and toxicity. The topic is important and the approach is novel. However, there are several limitations that, if addressed, would substantially improve the manuscript:

- 1) The logic regarding the role of mosaic mutations (discussion page 14) and the use of the arctic mutation should be made explicit in the abstract.
- 2) The fundamental limitation that toxicity to flies may not be similar to toxicity to humans should be noted as a limitation.
- 3) Amyloid-beta 1-42 with an arctic mutation is extraordinarily toxic. A key question is whether the platform would be sensitive to less severely toxic form of amyloid-beta. If not, it might not be broadly useful.
- 4) The possibility that some species other than amyloid-beta are being detected by the 6E10 antibody is a concern. 6E10 has incomplete specificity, and the bands on western blotting (Suppl Fig 1) are larger than would be expected for amyloid-beta.
- 5) (Minor) variant is misspelled on page 3
- 6) (Minor) An alternative interpretation of the immunodepletion experiments in ref 17 could be that another component associate with amyloid-beta could be responsible for seeding.

David Brody,
Washington University in St Louis

Reviewer #2 (Remarks to the Author):

The manuscript by Sowade and Jahn addresses the seeding hypothesis of Alzheimer's disease in an in vivo fly model. Although they made some interesting observations that support this hypothesis, there are several concerns.

- 1) They should provide a rational why the specific Gal4 lines werer chosen and provide more information in which cells they are active. It is also surprising that they do detect Aeta40 in the same areas addressed by these drivers when expressing it pan-neuronally (Fig. 2C). So they seemed to have pre-selected for expression in the same cell types. It would be more convincing if they would express the seed and the target in two clearly separated cell populations and then test whether aggregates spread through the brain and occur in the cells with the target.
- 2) Although the image in fig. 1 does not fit well with the expression pattern shown fro Abeta42. This may be because this is a staining of fibers and not cell bodies. But if this is the case a staining for cell bodies should be provided.
- 3) Although they do show that the levels of Abeta are important (as have other groups), the increase in insoluble Abeta in fig 2a and b does not correlate with the levels of Abeta42 seed. Althoughthey insoluble Abeta42 was below the detection levels in b (corresponding with the more restricted pattern), the combined effect with Abeta40 resulted in levels of insoluble Abeta that were twice as high as in a, although the expression in the optic lobes or Abeta42 alöone did result in detectable levels of insoluble Abeta.
- 4) Comparing the 6E10 staining with the FTAA staining (Fig. 3 and 4) indicates that there is a strong increase in soluble Abeta. However, this is not addressed in the manuscript. If there is a general increase in Abeta this would also cause more aggregates, independent from a seeding effect.
- 5) The stainings in figure 3 suggest that the aggregates occur within the cell bodies which should be confirmed by a cytoplasmic marker.

Minor comments: That Abeta levels correlate with toxicity has already been shown by other groups (e.g. Crowther et al., 2004), so the results in fig. 5 are more confirmative than novel.

Reviewer #3 (Remarks to the Author):

The manuscript by Sowade and Jahn describes a novel *Drosophila* model that allows for the characterization of Abeta 40 aggregation induced by locally restricted expression of Abeta 42 seeds. Contrary to previous spreading models, the aggregating protein is expressed localized in defined neuronal clusters. Localized expression of Abeta 42 induced widespread Abeta 40 accumulation over time, strongly suggesting that Abeta 42 directly seeded aggregation of Abeta 40. Importantly, the simultaneous localized expression of Abeta 42 and pan-neuronal Abeta 40 also significantly reduced life expectancy. As such, this study is original and will be of high interest to the community. This is a very elegant model and well suited to study mechanisms of non-autonomous protein aggregate induction. The manuscript is nicely written and experiments are carefully carried out.

My major concern is that a direct seeding of Abeta 40 through Abeta 42 aggregates has not been firmly demonstrated. General changes in proteostasis in the defined neuronal cluster could poise the environment and thereby induce misfolding of Abeta 40. A good control could be localized expression of another aggregation-prone protein, such as the Htt aminoterminal fragment with an expanded polyQ tract. (Babcock and Ganetzky, PNAS 2015).

Authors demonstrate a significant decrease in the survival rate in flies expressing both seed and target. Are other signs of neurotoxicity evident?

It is unclear if Abeta 42 seeds actually spread to distant brain regions or rather initiate a self-perpetuating Abeta 40 aggregation. Is it possible to perform Abeta 42 ELISA on isolated optic lobes when the seed is expressed centrally?

Page 10 and discussion: The concept of secondary nucleation through gene dosage increase is not well explained. It is unclear how exactly the presented model of Abeta 40 overexpression supports this concept.

Figure 2a, b, Suppl. Figure 1: The presence of insoluble Abeta in wildtype controls is unclear and likely related to background in western blots. The expression of Abeta 40 appears to decrease over time. Please explain.

Figure 2d: Legend for blue bars is misleading, as these bars represent insoluble Abeta 40 and not Abeta 40 AND Abeta 42.

Figure 5a: Is the difference between insoluble Abeta 40 in pan-neuronal Abeta 40 and Abeta40/Abeta 42 flies significant here?

Point by point responses to the reviewer comments

We are delighted to see the reviewers acknowledging our novel approach to tackle this crucial pathobiological concept. We are also thankful for the constructive criticism and comments, as they were very helpful in terms of improving the clarity of the manuscript. We have addressed all questions in detail and feel that the additional data sets and descriptive amendments have made the manuscript considerably stronger.

Reviewers' comments:

Reviewer #1 (Remarks to the Author):

This is an interesting report which provides proof of concept data for a drosophila model system for studying the relationship between amyloid-beta seeding and toxicity. The topic is important and the approach is novel. However, there are several limitations that, if addressed, would substantially improve the manuscript:

1) The logic regarding the role of mosaic mutations (discussion page 14) and the use of the arctic mutation should be made explicit in the abstract.

We adapted the abstract accordingly. Importantly, we also provide an additional set of data, where we have generated new models using the non-mutated A β ₄₂ peptide as seeding species. Here, we show that the observed effects in our models are not specific to the arctic mutation, but can also be initiated by non-mutated A β ₄₂ (see below and Supplementary figure 4).

2) The fundamental limitation that toxicity to flies may not be similar to toxicity to humans should be noted as a limitation.

While the fruit fly has been instrumental in deciphering some of the most complex biological systems, we agree that insight from model systems may only mimic certain aspects of the human disease mechanism. We have added this note into the discussion.

3) Amyloid-beta 1-42 with an arctic mutation is extraordinarily toxic. A key question is whether the platform would be sensitive to less severely toxic form of amyloid-beta. If not, it might not be broadly useful.

To directly address this aspect, we generated a new set of transgenic flies and performed a new set of experiments. Here, we examined how the amount of insoluble A β and the toxicity level change when the non-mutated, less-toxic A β ₄₂ (without the arctic mutation) is used as a seed. The results are added to the manuscript and presented in Supplementary figure 4. Notably, also this less toxic and less aggregation-prone A β variant in combination with one or two copies of the target peptide induced accelerated deposition of A β ₄₀. This increased A β deposition correlated with neurotoxicity, which is reflected in reduced survival of the flies as well as decreased locomotor activity. This is a strong confirmation of our initial findings and proves our model to be also suitable to study seeding ability of less toxic and less aggregation-prone peptide variants. This new dataset therefore strongly suggest that the introduced *Drosophila* models will be broadly useful for the research community.

4) The possibility that some species other than amyloid-beta are being detected by the

6E10 antibody is a concern. 6E10 has incomplete specificity, and the bands on western blotting (Suppl Fig 1) are larger than would be expected for amyloid-beta.

We agree with the referee that the 6E10 antibody is not completely specific in western blots, as we also find unspecific bands. However, we therefore always included the required controls (e.g. the driver line only) to check for the specificity of our signal and to exclude the analysis of unspecific bands. The here-quantified band, running just below 6 kDa in our *Drosophila* extracts, is exclusively observed in flies expressing A β . In addition, we have used a complementary technique to quantify our A β peptide levels. The increased A β deposition can be reproduced reliably using this electrochemiluminescent detection assay (ECL), which shows a higher sensitivity compared to western blots and is widely used within the A β community.

5) (Minor) variant is misspelled on page 3

This mistake was corrected.

6) (Minor) An alternative interpretation of the immunodepletion experiments in ref 17 could be that another component associate with amyloid-beta could be responsible for seeding.

The work described in ref. 17 elutes to this possibility. We have adjusted the sentence accordingly to reflect this.

David Brody,
Washington University in St Louis

Reviewer #2 (Remarks to the Author):

The manuscript by Sowade and Jahn addresses the seeding hypothesis of Alzheimer's disease in an in vivo fly model. Although they made some interesting observations that support this hypothesis, there are several concerns.

1) They should provide a rational why the specific Gal4 lines were chosen and provide more information in which cells they are active. It is also surprising that they do detect Aeta40 in the same areas addressed by these drivers when expressing it pan-neuronally (Fig. 2C). So they seemed to have pre-selected for expression in the same cell types. It would be more convincing if they would express the seed and the target in two clearly separated cell populations and then test whether aggregates spread through the brain and occur in the cells with the target.

The reviewer brings up an important point, which is the observation that the pathology of some neurotoxic proteins spreads along neuronal connections. This is less pronounced for the extracellular A β peptide. Here, our aim was to analyze A β seeding mechanisms and propagation of pathology. This does not necessarily imply that A β seeding-competent particles have to move via neuronal connections throughout the brain, but rather require their capability to induce the aggregation of the otherwise soluble cognate peptide in close proximity, resulting in a larger amount of seeding-competent material being produced and triggering a gradual seeding mechanisms throughout the brain.

For restricted seed expression (i.e. to trigger locally restricted seeding), we targeted 2 distinct neuronal clusters that differ in the cerebral area and in the number of involved neurons. The area targeted by *R9D03*-Gal4 lies in the optic lobes and comprises numerous cells, whereas *GMR14B06*-Gal4 only induces expression in a few neurons in the central

brain. Consequently, we were able to analyze whether the amount of seeds and the region of seed-expression determines the course of the seeding effect. The R9D03-driver represents a fragment of the earmuff gene, which encodes for an evolutionary conserved transcription factor (Fezl in vertebrates). Further characteristics of the targeted neurons of R9D03- and GMR14B06-Gal4 are not known. We chose the pan-neuronal driver *nSyb*-Gal4 for the pan-neuronal expression of the target peptide. Its overlap with the Gal4 drivers does not conflict with our conclusion, but actually mimics the physiological scenario, where throughout the brain A β ₄₀ and A β ₄₂ are produced by all neurons. However, local changes in neuronal protein homeostasis (selective vulnerability) or somatic point mutations may lead to the accumulation of an aggregation competent A β ₄₂ variant. We confirm this recently emerging concept by providing the first non-invasive *in vivo* data suggesting that very restricted changes in A β ₄₂ levels can trigger the A β seeding cascade. We have clarified this aspect in the main text.

2) Although the image in fig. 1 does not fit well with the expression pattern shown for Abeta42. This may be because this is a staining of fibers and not cell bodies. But if this is the case a staining for cell bodies should be provided.

This point is addressed below in combination with point 5.

3) Although they do show that the levels of Abeta are important (as have other groups), the increase in insoluble Abeta in fig 2a and b does not correlate with the levels of Abeta42 seed. Although the insoluble Abeta42 was below the detection levels in b (corresponding with the more restricted pattern), the combined effect with Abeta40 resulted in levels of insoluble Abeta that were twice as high as in a, although the expression in the optic lobes or Abeta42 alone did result in detectable levels of insoluble Abeta.

The reviewer points towards an important aspect of our study that required more detailed discussion in the manuscript. Aim of this work is not to show that overexpressing increasing amounts of aggregation-prone A β results in increasingly toxic effects. We agree, his correlation has been described previously in numerous over-expression models from yeast to rodents. In contrast, our seeding models demonstrate that the local expression of minor amounts of aggregation-prone A β ₄₂ (seed), at levels hardly detectable by sensitive ECL methods, can elicit a seeding cascade resulting in increased deposition of an A β variant otherwise not prone to aggregation when overexpressed at high levels. We have re-analyzed the A β levels in our models and present the comparison between total and insoluble A β levels in Supplementary figure 2. In this respect, it is striking that lower amounts of the seed in the central brain induce an apparently stronger seeding effect than larger amounts that are present in the optic lobes. These findings support the importance of the neuronal environment and we envision that future studies, using the herein described experimental setup, will target this specific question. In addition, we now provide a new set of data where we use the non-mutated A β ₄₂ variant for our seeding studies (Supplementary figure 4). This variant is less prone to aggregation and has been described to aggregate into fibrillar structures with distinct conformational characteristic, sometimes referred to as strains. These initial data suggest that also the conformational characteristics of these seeds will determine the seeding efficacy and subsequent propagation processes. These results are novel and while we do not have the complete mechanistic insights, the impact of this study is high. We have made corresponding changes in the manuscript to further elucidate on these findings.

4) Comparing the 6E10 staining with the FTAA staining (Fig. 3 and 4) indicates that there is a strong increase in soluble Abeta. However, this is not addressed in the manuscript. If there is a general increase in Abeta this would also cause more aggregates, independent from a seeding effect.

We agree with the reviewer comment that the induction of aggregation by a particular A β species, rather than an increase in the total amount of any A β species, would be in line with the seeding hypothesis. We have specifically addressed this aspect in our work, by measured insoluble and total A β_{40} levels in flies expressing the seed and two copies of the target peptide. As a control we used flies that expressed A β_{40} instead of A β_{42} arctic as the seed. The results are presented in Supplementary Figure 2. Similar levels of total A β are reached independent from the type of seed (A β_{40} or A β_{42} arctic). However, when comparing the level of insoluble A β_{40} , it becomes apparent that A β_{42} arctic, but not A β_{40} peptides, induce an increase in the amount of insoluble A β_{40} . Thus, not the level of total A β determine the amount of insoluble A β , but the presence of aggregation prone A β_{42} seeds induce a seeding event that results in enhanced deposition of the otherwise-soluble A β_{40} . We have also added a section into the manuscript to describe these results and discuss this finding.

2) Although the image in fig. 1 does not fit well with the expression pattern shown for Abeta42. This may be because this is a staining of fibers and not cell bodies. But if this is the case a staining for cell bodies should be provided.

5) The stainings in figure 3 suggest that the aggregates occur within the cell bodies which should be confirmed by a cytoplasmic marker.

In the human brain A β has been found extra- as well as intracellularly and so far it is not known which A β pool is responsible for the seeding mechanisms. Here, it is not clear by what mechanism and to what level A β is taken up by neurons. Analyzing this quantitatively in our *Drosophila* models has been very challenging, given the small size of neuronal cell bodies and the lack of sensitive antibodies for extracellular markers. Also the 6E10/pFTAA staining expectedly shows a different pattern than intracellularly expressed mCherry, we have not been able to identify conditions to clearly visualize the extracellular localization of A β seeds. However, while others and we have previously shown the efficacy of our secretion signal peptide using *Drosophila* cell lines, it remains challenging to follow secretion *in vivo*. However, to address the reviewer comment, we have now generated a *Drosophila* line expressing A β_{42} in the absence of a secretion peptide. With this line we aimed to address the cellular localization and intracellular abundance of A β . Importantly, A β accumulation can only be detected when A β_{42} carries the signal peptide, that is, when it is targeted for secretion (see additional figure A1 below). A β_{42} with no secretion peptide (A β_{42} NSP) is efficiently degraded inside cells. These data support the requirement for secretion of A β in our *Drosophila* models and suggest that A β accumulations mask the outside of the cells.

Figure to Reviewers A1 | Confocal images of antennal lobes of flies expressing $A\beta_{42}$ with (a) or without (non-signal peptide, NSP, b) the signal peptide. $A\beta$ -expression was driven in the central brain using *GMR14B06*-Gal4 and brains were stained with the monoclonal $A\beta$ antibody 6E10. $A\beta$ accumulation can only be observed when the signal peptide is included, i.e. when $A\beta$ is targeted for secretion, whereas $A\beta_{42}$ lacking the signal peptide (NSP) cannot be detected in the target region.

Minor comments: That Abeta levels correlate with toxicity has already been shown by other groups (e.g. Crowther et al., 2004), so the results in fig. 5 are more confirmative than novel.

We respectfully disagree with this comment. Whereas the correlation between overall levels of pathogenic $A\beta$ species and neurotoxicity has been shown before, the novelty of our study is that we demonstrate that very low amounts of $A\beta_{42}$ seeds, that themselves are not toxic, are sufficient to induce the accelerated deposition of $A\beta_{40}$ which correlates with a severe survival phenotype. In the “double-target” experiment the deposition of $A\beta_{40}$ is further induced (accompanied by only a slight increase in the amount of insoluble $A\beta_{42}$), which correlates with an aggravated survival phenotype. The results show that the otherwise soluble and non-toxic $A\beta_{40}$ is driven to form toxic accumulations, in the presence of these potent peptide seeds. This link between endogenous seeding mechanisms and severe neurotoxicity is novel, as it is distinct from classic overexpression models in various animal species and distinct from invasive seeding models recently described in rodents.

Reviewer #3 (Remarks to the Author):

The manuscript by Sowade and Jahn describes a novel Drosophila model that allows for the characterization of Abeta 40 aggregation induced by locally restricted expression of Abeta 42 seeds. Contrary to previous spreading models, the aggregating protein is expressed localized in defined neuronal clusters. Localized expression of Abeta 42 induced widespread Abeta 40 accumulation over time, strongly suggesting that Abeta 42 directly seeded aggregation of Abeta 40. Importantly, the simultaneous localized expression of

Abeta 42 and pan-neuronal Abeta 40 also significantly reduced life expectancy. As such, this study is original and will be of high interest to the community. This is a very elegant model and well suited to study mechanisms of non-autonomous protein aggregate induction. The manuscript is nicely written and experiments are carefully carried out.

My major concern is that a direct seeding of Abeta 40 through Abeta 42 aggregates has not been firmly demonstrated. General changes in proteostasis in the defined neuronal cluster could poise the environment and thereby induce misfolding of Abeta 40. A good control could be localized expression of another aggregation-prone protein, such as the Htt aminoterminal fragment with an expanded polyQ tract. (Babcock and Ganetzky, PNAS 2015).

We thank the reviewer for bringing up this interesting point. We fully agree that our novel *Drosophila* models are amenable to address this intriguing question of neuronal proteostasis and selective neuronal vulnerability. This line of research will be part of future studies by many different laboratories. To address this reviewer question, we generated new transgenic fly lines where we introduced aggregation-prone Huntingtin seeds with an expanded polyQ tract (HttQ72) in combination with the target peptide A β ₄₀. Seed expression was driven in the central brain using *GMR14B06*-Gal4. The HttQ72 construct is eGFP-tagged, allowing us to confirm HttQ72 expression via immunohistochemistry. Subsequently, we analyzed levels of insoluble A β ₄₀ in the presence of either A β ₄₂arctic or HttQ72 seeds, to perform a direct comparison in this new genetic setup. The results are shown below in additional figure A2. Importantly, HttQ72 seeds did not lead to an increase in the amount of insoluble A β . In contrast, we could detect an increase in the amount of insoluble A β when using A β ₄₂arctic seeds in combination with the target peptide (of note, the lower levels of insoluble A β in comparison the main manuscript data are based on reduced transgene expression in the required transgenic lines). These preliminary results suggest that the seeding effect that we have observed is not simply caused by general alterations in proteostasis, but that it is specifically induced by fast aggregating A β variants. A further support for this hypothesis is provided within the new Supplementary figure 4. There, we show that the less aggregation-prone and less toxic A β ₄₂, therefore perturbing proteostasis in the cell to a smaller extent than A β ₄₂arctic, induces the deposition of A β ₄₀ to an even higher degree. We believe that extending these *in vivo* studies further to other possible proteostasis regulatory mechanisms is very intriguing, but beyond the scope of this manuscript. Thus, although our data collectively suggest that direct seeding mechanism drive the deposition of A β pathology, the specific neurodegenerative mechanisms remain to be formally studied.

Figure to Reviewers A2 | Huntingtin seeds do not induce accelerated deposition of Aβ. ECL-measurement of insoluble Aβ in 21-day-old flies expressing either Aβ₄₂arctic or HttQ72 seeds in the central brain using GMR14B06-Gal4 in addition to the pan-neuronally expressed Aβ₄₀. **(a)** The amount of insoluble total Aβ reflects the sum of insoluble Aβ₄₀ and Aβ₄₂. Significances were determined using one-way ANOVA in comparison with flies expressing target only (error bars, s.e.m.; n=5 independent biological replicates; *P ≤ 0.05). **(b)** Relative amount of insoluble Aβ₄₀ normalized to “target only” (grey column). Significance was determined using one-way ANOVA (error bars, s.e.m.; n=4 independent biological replicates; *P ≤ 0.05).

Authors demonstrate a significant decrease in the survival rate in flies expressing both seed and target. Are other signs of neurotoxicity evident?

This is a very interesting question in regard to the translatability of our model. We performed additional immunohistochemistry experiments, however we could not detect changes in cleaved caspase staining (sign for apoptosis, antibody: Cleaved *Drosophila* Dcp-1 (Asp216) by Cell Signaling Technologies) or gamma-H2Av staining (marker for DNA damage, antibody: Histone H2AvD pS137 by BioTrend). As a consequence, we decided to setup locomotor measurements as an additional phenotypic readout of neuronal integrity, which is a well described measure for invertebrate as well as rodent neurodegeneration models. Here, we analyzed the locomotion of flies using a semi-automated camera setup. Importantly, we were able to closely replicate the effect of Aβ₄₂ seeding on neurodegeneration, with locomotion being the more direct measure of neurotoxicity, observable well before the impact on fly survival. These data are collectively presented in Supplementary figure 4 and discussed in the main manuscript.

It is unclear if Abeta 42 seeds actually spread to distant brain regions or rather initiate a self-perpetuating Abeta 40 aggregation. Is it possible to perform Abeta 42 ELISA on isolated optic lobes when the seed is expressed centrally?

This is a very interesting approach, which however turns out to be very challenging. In general it might be possible to do ELISA measurements with isolated optic lobes, however, the reproducible isolation and required tissue manipulations to isolate sufficient material from fly brains was technically too variable. Having a closer look at brains expressing the seed in the central brain, we did not observe any p-FTAA positive accumulations outside the target region of the GMR14B06 driver. This suggests that either the seeds are not traveling throughout the brain or that the detection method is not sensitive enough to visualize them. Based on these studies, we are not able to disregard the possibility that Aβ seeds spread

independently through the brain tissue. However, our data suggest a very strong dependence on the availability of target peptide and the seeding-process itself to be main drivers for this progression. We have added an additional point to the discussion.

Page 10 and discussion: The concept of secondary nucleation through gene dosage increase is not well explained. It is unclear how exactly the presented model of Abeta 40 overexpression supports this concept.

We thank the reviewer for pointing out this unclarity. While the initial rate-limiting step of *de novo* aggregation is the generation of a nucleating species, the propagation of aggregation is mainly determined by the amount of free peptide available for conversion. Here, we have specifically introduced seeds into the A β ₄₀ background by expressing the aggregation-competent A β ₄₂ species. Therefore we hypothesized, and experimentally observed, that the amount of A β ₄₀ available for the templated aggregation reaction is the main determinant of the amount of fibrils formed, rather than the level of the initial seed. We adapted the paragraphs in order to make this point more clear.

Figure 2a, b, Suppl. Figure 1: The presence of insoluble Abeta in wildtype controls is unclear and likely related to background in western blots. The expression of Abeta 40 appears to decrease over time. Please explain.

We thank the reviewer for pointing out the lack of explanation. The apparent A β levels are indeed based on experimental background. This point has been added into the main text. Furthermore, the apparent reduction in the accumulation of A β ₄₀ with increasing lifespan can be explained by two effects. First, the soluble A β ₄₀ is cleared efficiently from the fly brain in the case where no aggregation is induced. In addition, we also observed a reduction in the expression-efficacy of the used pan-neuronal LexA driver with increasing lifespan. These two factors result in the apparent decrease of A β ₄₀ with time. We also added this point into the main text.

Figure 2d: Legend for blue bars is misleading, as these bars represent insoluble Abeta 40 and not Abeta 40 AND Abeta 42.

We have changed the legend accordingly to clarify.

Figure 5a: Is the difference between insoluble Abeta 40 in pan-neuronal Abeta 40 and Abeta40/ Abeta 42 flies significant here?

The levels of insoluble A β ₄₀ are significantly increased in flies also expressing the A β ₄₂arctic variant. For clarity, we have normalized the data in Figure 5 to the levels of flies expressing a single copy of A β ₄₀ for the entire data set. Therefore, we are not plotting the individual significances, which we have stated in the Figure legend accordingly. Significances for the direct comparison are indicated in Figure 2.

REVIEWERS' COMMENTS:

Reviewer #1 (Remarks to the Author):

The authors have nicely revised the manuscript and addressed my critiques.
DB

Reviewer #2 (Remarks to the Author):

The authors have addressed my previous concerns and added additional experiments as well as clarified points raised.

Reviewer #3 (Remarks to the Author):

The authors have adequately addressed my concerns.